# Digital health interventions for gestational diabetes mellitus: A systematic review and meta-analysis of randomised controlled trials

**Boutheina Leblalta[1], Hanane Kebaili[2], Ruth Sim[3], Shaun Wen Huey Lee[3,4,5]**\*

**1** Faculty of Medicine, Department of Pharmacy, University of Salah Boubnider Constantine, Algeria, **2** Faculty of Médecine, Département of Pharmacy, Benboulaid University of Batna, Algeria, **3** School of Pharmacy, Monash University Malaysia, Jalan Lagoon Selatan, Selangor, Malaysia, **4** School of Pharmacy, Faculty of Health and Medical Sciences, Taylor's University, Subang Jaya, Selangor, Malaysia, **5** Center for Public Health, University of Pennsylvania, Philadelphia, Pennsylvania, United States of America

\* shaun.lee@monash.edu

**Data Availability Statement:** All data are presented in the manuscript.

**Funding:** The authors received no specific funding for this work.

## Abstract

Good blood glucose control is important to reduce the risk of adverse effects on mothers and their offspring in women with gestational diabetes (GDM). This review examined the impact of using digital health interventions on reported glycaemic control among pregnant women with GDM and its impact on maternal and foetal outcomes. Seven databases were searched from database inception to October 31st, 2021 for randomised controlled trials that examined digital health interventions to provide services remotely for women with GDM. Two authors independently screened and assessed the studies for eligibility for inclusion. Risk of bias was independently assessed using the Cochrane Collaboration's tool. Studies were pooled using random effects model and presented as risk ratio or mean difference with 95% confidence intervals. Quality of evidence was assessed using GRADE framework. Twenty-eight randomised controlled trials that examined digital health interventions in 3,228 pregnant women with GDM were included. Moderate certainty of evidence showed that digital health interventions improved glycaemic control among pregnant women, with lower fasting plasma glucose (mean difference -0.33 mmol/L; 95% CI: -0.59 to -0.07), 2-hour postprandial glucose (-0.49 mmol/L; -0.83 to -0.15) and HbA1c (-0.36%; -0.65 to -0.07). Among those randomised to digital health interventions, there was a lower need for caesarean delivery (Relative risk: 0.81; 0.69 to 0.95; high certainty) and foetal macrosomia (0.67; 0.48 to 0.95; high certainty). Other maternal and foetal outcomes were not significantly different between both groups. Moderate to high certainty evidence support the use of digital health interventions, as these appear to improve glycaemic control and reduce the need for caesarean delivery. However, more robust evidence is needed before it can be offered as a choice to supplement or replace clinic follow up.

**Systematic review registration**: PROSPERO: CRD42016043009.

**Competing interests:** The authors have declared that no competing interests exist.

## Author summary

Gestational diabetes is the most common medical complication of pregnancy, affecting between 1% to 45% of pregnancies depending on population and diagnostic criteria. Treatment primarily focuses on counselling, dietary modification and increasing physical activity. Optimal management also requires the mothers' involvement to self-monitor blood glucose levels. Digital health interventions such as smartphone apps, SMS messaging and websites can provide behavioural support and education needs of these mothers. In this systematic review and meta-analysis, we assessed the use of digital health interventions to support the management of mothers with gestational diabetes and its impact on glycaemic control. We found that the use of digital health interventions were associated with better glucose control and lower weight gains over pregnancy, which reduces the risk of complications for both baby and mother during delivery. We also found that mothers with gestational diabetes had lower need for caesarean delivery while their babies were more likely to be born within the recommended weight range. Nevertheless, until more robust evidence is found, digital health should only be used as an adjunct in addition to regular clinic follow-up.

## Introduction

Gestational diabetes mellitus (GDM) is an increasingly common diagnosis during pregnancy [1] and has a substantial effect on maternal and foetal morbidity including risk of developing preeclampsia, shoulder dystocia, caesarean delivery [2] as well as future health complications such as cardiovascular diseases, type 2 diabetes and cancer risk. [3] The prevalence of GDM was estimated to be 8% to 9% of all pregnancies, [4] and is rising due to an increased rate of obesity, changing threshold of GDM as well as lifestyle changes such as physical inactivity and the adoption of modern lifestyles. [5,6] Women with GDM have a significant increase lifetime risk of developing type 2 diabetes, and a three-fold increase in developing metabolic syndrome and cardio-vascular diseases. [7,8] Current guidelines have recommended the need for post-partum follow up and care, including continued support for lifestyle changes. [9,10] In women with GDM, medical nutrition therapy remains the mainstay treatment with daily self-monitoring of blood glucose (SMBG), aimed at normalizing blood glucose to reduce the risk of complications as well as improve maternal and foetal outcomes as well as risk for developing metabolic syndromes in the future. [11]

Technological innovations have provided opportunities for novel approaches to improve the care of people with diabetes and women with GDM. [12–14] Some potential benefits of using technology include the ability to provide support and immediate feedback, reducing the distance barriers as well as reduce healthcare costs through resource pooling. [15] Existing literature reviews performed to date on digital health interventions, defined as electronic systems in medicine and other health professions designed to provide services remotely to manage illness and health risk and promote wellness is key towards universal health coverage, as it provides safe, timely and affordable access to health services for all. [16–18] The premise is that digital health interventions can facilitate training, surveillance, and service delivery, and more importantly empower users of digital health to make better informed decisions about their own health in new and innovative ways. [19–22] Some examples of digital health interventions used in diabetes care include telehealth, game-based support, mobile health (mHealth) as well as patient portals. [12,17] Digital health interventions can be an important solution especially for women with GDM, given the limited time clinicians have to manage and educate these

women who requires short-term adjustment to their therapy. More importantly, digital health interventions can help ensure that health resources are optimally utilised. [15]

Several reviews have suggested that there were limited benefits of telemedicine in women with GDM. [23,24] In one of the earliest systematic review on the use of telemedicine for people with GDM in 2015, Rasekaba *et al* in their meta-analysis of three studies reported that telemedicine had no beneficial impact on maternal outcomes. [23] However, an updated review by Ming and colleagues of seven randomised controlled studies reported that telemedicine was useful to reduce HbA1c but not maternal and neonatal outcomes. [24] Garg *et al* in their recent review in 2020 attempted to clarify this association, and reported that mobile apps along with medical care could be effective to manage and prevent risk associated with GDM. [25] Since the publication of these reviews, several new trials have been conducted which have reported otherwise. Guo and colleagues recently conducted a randomised controlled study among 124 women with GDM and noted that digital health was effective in reducing maternal blood glucose compared to control. [26] Yew *et al* similarly reported that the introduction of a smartphone based monitoring platform was effective in reducing maternal blood glucose as well as lower rate of pregnancies requiring insulin treatment compared to controls. [27] In light of these new developments and to address these important knowledge gaps, we conducted a systematic review and meta-analysis to investigate the efficacy of digital health interventions use to support women with GDM.

## Materials and methods

### Sources

We searched for studies examining the use of digital health interventions in woman with gestational diabetes on the following databases: Cochrane Library, EMBASE, Ovid MEDLINE, CINAHL Plus, Maternity & Infant Care database, and PsycINFO from database inception to March, 31st 2021 without any language restriction using the search terms as listed in the S1 Appendix. This was supplemented by a hand search of the reference list of retrieved articles, the CNKI database and relevant systematic reviews. We also searched ClinicalTrials.gov and the World Health Organization's International Clinical Trial Registry Platform to identify for any additional on-going or unpublished studies using the search term "gestational diabetes", as trial registries can be an important source to identify for additional studies. [28] We updated the search results upon peer review to include studies up to October 31st, 2021

This study adhered to the 2020 Preferred Reporting Items for Systematic Review and Meta-Analyses for RCTs [29,30], and the protocol was registered (PROSPERO Identifier: CRD42016043009).

### Study selection and eligibility criteria

Studies were included in the current review if they were: 1) randomised controlled trials (RCTs); 2) conducted in pregnant women with gestational diabetes; and 3) examined the use of digital health intervention, defined as the use of technology such as telephone, mobile phone, video-conferencing or web-based interface for medical exchange of health related information. [31] Studies were excluded if they had examined women with pre-existing type 1 or type 2 diabetes. Two authors (BL, HK or RS) independently screened the titles and abstracts, and retrieved the full text of any articles considered eligible. Any disagreements were resolved through consensus or adjudication by a third reviewer (SWHL).

### Data extraction and quality assessment

Data on baseline characteristics and intervention details, maternal and neonatal outcomes measures such as blood glucose, complication rates were independently extracted by two

reviewers (BL, HK or RS). All data were double-checked by a third reviewer (SWHL) for accuracy before analysis. When a study reported both GDM and type 1 or 2 diabetes, only data for GDM were extracted and included. If data were not available in numerical format, we estimated it from the figures using WebPlotDigitizer. [32] The methodological quality of all included studies were assessed using the updated Cochrane Risk of Bias 2.0 tool. [19]

## Outcomes

The primary outcome of interest was the reported glycaemic control among women with GDM, in particular mean fasting glucose, post-prandial glucose and glycated haemoglobin ($HbA_{1c}$). Other outcomes of interest include markers of adequate glycaemic control such as weight gain during pregnancy, number requiring caesarean delivery, medication use, macrosomia, large for gestational age, number of neonates requiring intensive unit care. We also reported patient outcomes such as quality of life, health care cost such as number of scheduled clinic visits and cost-effectiveness.

## Intervention classification

Intervention and control conditions were subsequently classified into either one of the following categories based upon an adaptation of the definitions from the American Telemedicine Association (2016) reported previously by Lee et al. [13]

- Tele-education: Any intervention aimed at educating teaching, or training patients remotely using live interactive streaming or by stored educational material.

- Telemonitoring: Any process which allows for the delivery and/or exchange of information to monitor a health status of patients remotely.

The classification of intervention was based upon the primary aim of each trial, such as to address the lack of monitoring or to provide health related education. For example, if the digital health intervention targets to educate a patient on the importance of blood glucose monitoring but does not allow for exchange of information or feedback, this was classified as tele-education. In comparison, if the trial request that participants monitor their blood glucose levels and provides feedback based upon these readings, the intervention was classified as a telemonitoring study.

As the technologies used varied within each study, we followed the approach taken by previous reviews and classified these as one of the following:- telehealth, mobile health or mHealth (where mobile devices are used to support medical care), game-based support, social platforms and patients portal (see Table 1 for full definitions). [12,33]

## Data synthesis and analysis

All data were summarized and presented narratively. In studies that had reported similar outcomes, data were pooled and presented as mean difference and their 95% confidence intervals (95% CI), calculated either end of treatment values or change from baseline values. For binary outcomes, these were presented as risk ratio with the 95% CIs. Meta-analysis were performed using the DerSimonian-Laird method, since clinical and methodological heterogeneity was likely to exists and have an effect on the results. We also stratified the different types of intervention based upon classification described as above as well as digital health interventions used. The Cochran Q test and $I^2$ statistics were used to examine statistical heterogeneity. Publication bias was assessed visually to check for funnel plot asymmetry and if an outcome measure had 5 or more studies, Egger's test was performed. As a priori, we planned several

**Table 1. Definition of digital health interventions used in this study.**

**Telehealth**

Telehealth refers to the use of electronic medium (e.g. videoconferencing, telephone calls) which facilitates synchronous (real-time) communication between a patient and healthcare provider. The aim of such technology is to reduce the geographical barrier between both individual without sacrificing access to tailored treatment and live interaction.

**mHealth**

mHealth can be subdivided into 2 distinct technologies namely messaging systems and mobile applications.
*Messaging systems*
Messaging systems technologies include short message service (SMS), text messaging, and email. These are asynchronous communications, do not include an auditory component, and typically are unlikely to be personalised to the individual's need.
*Mobile applications*
Applications (apps) or software downloaded from a website or an app store and accessible via smartphones and tablet devices. The software is designed to fulfil a particular purpose, which can include self-management education, psychoeducation, reference sources (eg, database of nutritional content of foods), data tracking (e.g., physical activity, diet, blood glucose levels), and behavioural interventions.

**Game-based support**

Computer and video games which have been developed to facilitate diabetes education and promote self-management. Typically, they include situational problem-solving and interactive activities and reinforce health behaviours to improve diabetes outcomes.
They are largely targeted at children, adolescents, and young adults with type 1 diabetes.

**Social platforms**

Web-based social platforms enable people with diabetes to access social support without geographic boundaries, forming online health communities. These platforms include widely used social media sites (eg, Facebook, Twitter), and various other discussion forums. Web-based social platforms create unique opportunities for online peer support as well as diabetes education and intervention.

**Patient portals**

Online interactive treatment environments are systems that facilitate sharing of personal health records between the individual with type 1 or 2 diabetes and their health professionals and provide multiple methods for self-managing health information. Portal functions can include online appointment scheduling, appointment reminders, prescription refill requests, journaling and tracking tools, opportunities for health professional support, psychoeducational tools, and the ability to upload, view, and manage health information.

subgroup analyses to explore treatment association according to the following categories: 1) type of digital health; 2) intervention classification; 3) trials with large sample size of >100 participants; 4) study locations; and 5) GDM diagnosis criteria. Upon peer review, we also examined if results differed if only high-quality studies were included. The quality of each outcome reported was summarized using the Grading of Recommendations Assessment, Development and Evaluation (GRADE) criteria. [34] All analyses were performed on Review Manager 5.4.1 (Copenhagen: The Nordic Cochrane Centre, The Cochrane Collaboration).

# Results

## Study selection

The search on 7 databases identified a total of 521 articles, with 395 remaining after removal of duplicates. Of these, 54 articles were selected for further review after screening of titles. A total of 26 studies were excluded as they were either a non-randomised study, were abstracts or did not examine the population of interest. This review included twenty-eight studies (Fig 1).

## Characteristics of included studies

The included studies were published between 2007–2021, with more than half (n = 18; 64.3%) of studies published within the past five years. These studies were conducted in Asia (17 studies) [27,35–51], Europe (6 studies) [52–57], United States (3 studies) [58–60] and Australia (2

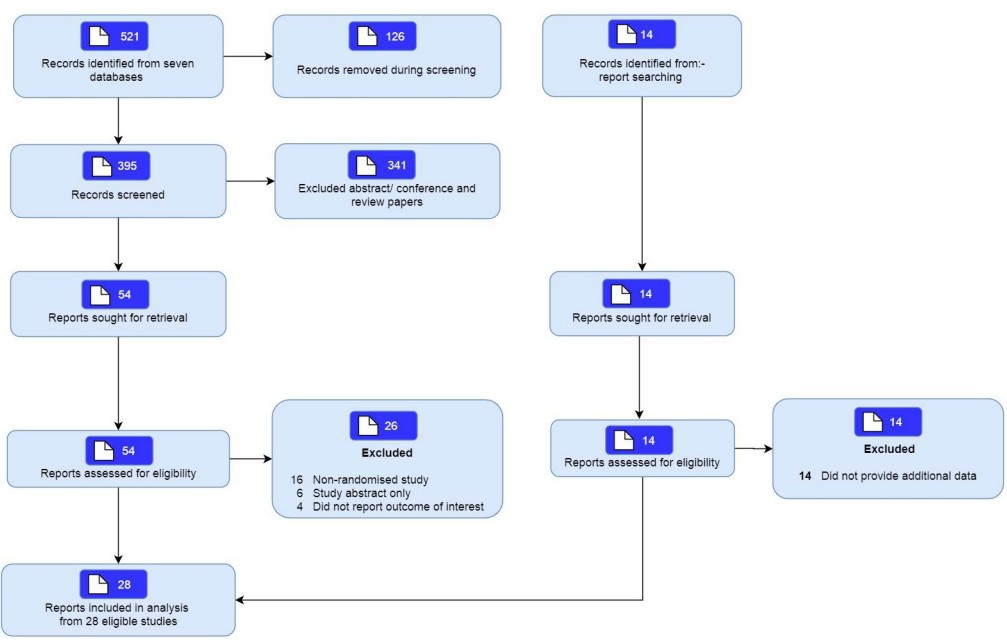

**Fig 1. Study flow.**

studies) [61,62]. Participants were pregnant women that were between 24 to 36 weeks of gestation at inclusion, with mean age between 25 to 39 years old. Seven studies reported the ethnicity of their participants. [27,52,53,55,58–60]. All studies were performed in hospital setting. The interventions included mHealth interventions (n = 18) [27,35–37,39,41–45,50–53,58], telehealth (n = 1) [38], patient portal (n = 8) [54–57,59–62] and social platforms (n = 4) [40,46–48]. Most of the trials were single centre studies and had small sample sizes. Only five trials included over 200 participants [27,50–53], four had 100–200 participants [37,40,42,62], and nineteen had fewer than 100 participants [35,36,38,39,41,43–48,54–61]. Only twenty-two studies had detailed the diagnostic criteria for GDM (Table 2).

## Assessment of intervention

There were methodological variations across the trials with regards to the intervention. Almost all studies were telemonitoring trials, whereby the participants were asked to monitor their blood glucose levels, which were stored and forwarded for review and feedback by the team remotely (asynchronous). The frequency of monitoring varied between studies and could range from once daily up to six times a day (Table 2). Four studies were tele-education trial, which compared the efficacy of delivering educational materials online compared to face-to-face session at the clinic. [38–40,62] Studies that were conducted before 2017 had used either websites, patient portals, telephone systems or short messaging system that allowed for exchange of blood glucose readings in their intervention. Mobile phones and health related apps were utilised in all except two studies [61,62] conducted after 2018.

In all trials, the comparison group usually received routine care, which differed in protocol depending on the location and practice but in all cases, SMBG was suggested albeit different monitoring frequencies. Eight studies [36,37,45,47,50,53,60] also reported participants adherence to study protocol. However, the definitions used varied depending on studies. This includes the total number of glucose monitoring performed as recommended in the study

**Table 2. Characteristics of included studies.**

| Author, Year (Country) | Participants characteristics* | Definition of GDM | Intervention component details | Monitoring frequency in study | Diabetes care team | Comparator |
|---|---|---|---|---|---|---|
| Yew et al, 2021 [27] (Singapore) | Age: 32.0 years, gestational age 26.9 weeks, BMI: 25.6kg/m$^2$, Total participants: 340 | 2013 World Health Organisation criteria | Habits-GDM app lifestyle coaching program that integrated dietary, physical activity, weight, stress and glucose monitoring advice. Participants sent wight and blood glucose results with a manual chat function with healthcare team response within 24 hours. | Seven times daily, for 2–3 days a week | Dietician, diabetic nurse educator, obstetricians, endocrinologist | Routine visits to antenatal clinics with group based education delivered face-to-face for 1.5 hours. Participants recorded SMBG on paper diary |
| Tian et al, 2021 [50] (China) | Age: 31.1 years, gestational age: 26.4 weeks, BMI: 24.0kg/m$^2$, Total participants: 309 | International Association of Diabetes and Pregnancy Study Group criteria | Social media app (WeChat) which patients are sent educational materials related to gestational diabetes. Researchers provided individualised coaching and feedback. | Five times a day for 6 days within a 2 week period | Obstetricians, nutritionists, nurses, health managers, psychologists, sports medicine | Routine care with a minimum of one educational counselling |
| Huang et al, 2021 [51] (China) | Age: 31.0 years, gestational age: 26.9 weeks, BMI: 25.0kg/m$^2$, Total participants: 295 | International Association of Diabetes and Pregnancy Study Group criteria | Social media app (WeChat) which patients are sent educational materials related to gestational diabetes. | NR | NR | Routine care as per clinic protocol |
| Huang et al, 2020 [46] (China) | Age: 30 years, gestational age 24–28 weeks, BMI: 24.35kg/m$^2$, Total participants: 88 | 2011 American Diabetes Association criteria | Social media app (WeChat) which patients are sent educational materials related to gestational diabetes. Platform acts as a channel for online peer support with ability for individualised coaching | NR | Nurse | Routine care with health education related to diabetes |
| Al-Ofi et al, 2019 [35] (Saudi Arabia) | Age: 32.4 years, gestational age 24–28 weeks, BMI: 30.6kg/m$^2$, Total participants: 57 | International Association of Diabetes and Pregnancy Study Group criteria | Telemonitoring device, comprising of a smartphone-glucometer and a Glucomail application to monitor blood glucose and weight. Monitoring until 6 weeks post-delivery. Information was sent and reviewed weekly by diabetic care team, for further interventions, such as lifestyle monitoring or insulin/medication adjustments. | Four times daily, in the morning before breakfast and three 2-hr post-prandial test | NR | Routine visits to antenatal clinics with referral to dietician if needed |
| Borgen et al, 2019 [52] (Norway) | Gestational age 24–28 weeks, Total participants: 238 | 2-hour OGTT $\geq 9$ mmol/l | Mobile health app (Pregnant+ app) to support in the management and monitoring, providing dietary and physical activity advice and feedback on diabetes related problem. The app also had links to educational resources. | NR | Midwife, diabetic nurse | Routine care as per clinic protocol, including SMBG monitoring, education, dietary and physical activity counselling |

(*Continued*)

**Table 2.** (Continued)

| Author, Year (Country) | Participants characteristics* | Definition of GDM | Intervention component details | Monitoring frequency in study | Diabetes care team | Comparator |
|---|---|---|---|---|---|---|
| Carolan-Olah et al, 2019 [62] (Australia) | Age: 31.7 years, gestational age 28–32 weeks, BMI: 30.2kg/m$^2$, Total participants: 110 | NR | Web-based educational intervention comprising of 4 modules on SMBG, diet, healthy habits and managing emotions. | Four times daily, in the morning before breakfast and three 2-hr post-prandial test | Dietician, diabetic educator | Routine visits to antenatal clinics with group based education delivered face-to-face for 1.5 hours |
| Cui et al, 2019 [45] (China) | Age: 29.5 years, gestational age 28.5 weeks, Total participants: 80 | 2011 American Diabetes Association criteria | Daily short-messaging system (SMS) reminders on advice related to diet, exercise and diabetes related problem. Weekly SMS were sent to remind patients to test for SMBG, with immediate feedback and consultation if needed | Weekly | NR | Routine care as per clinic protocol, including diet, exercise and diabetes care related advice |
| Guo et al,2019 [26] (China) | Age: 30.9 years, gestational age 24.9 weeks, BMI: 25.7 kg/m$^2$, Total participants: 60 | 2011 American Diabetes Association criteria | SMBG daily sent via mobile app (Dnurse) for review, with feedback sent daily. Feedback include dietary and physical activity advice, medication management and any diabetes related issues. The app also had links to educational resources. | Twice daily for 3 days of the week, fasting, and 2h-post-prandial reduced to twice a week once blood glucose stabilised. | Nurse, physician | Routine care as per clinic protocol, including SMBG as per intervention recorded on diary and dietary advice with clinic follow up |
| Sung et al, 2019 [44] (South Korea) | Age: 33.4 years, gestational age 27.3 weeks, BMI: 25.5 kg/m$^2$, Total participants: 21 | International Association of Diabetes and Pregnancy Study Group criteria | Mobile app which patients send SMBG for review, with feedback on medical and nutritional guidance sent twice weekly. Additional educational messages sent each week together in addition to standard care | Four times daily with food diary recording | Obstetrician a, nurse, nutritionist and endocrinologist | Biweekly visit up to 36 weeks of gestation followed by weekly visit until delivery |
| Yu et al, 2019 [48] (China) | Total participants: 90 | NR | Social media app (WeChat) which patients are monitored, and sent motivational educational materials related to gestational diabetes including videos and pictures daily. Daily forum setup to answer any related queries setup with gamification features. | NR | Gynaecologist, nurse, psychologist | Routine clinic care and health education. Patients also encouraged to attend consultation and monitor SMBG. |
| Jiang et al, 2018 [47] (China) | Age: 29.3 years, gestational age 25.3 weeks, Total participants: 80 | 2011 American Diabetes Association criteria | Social media app (WeChat) which patients are monitored, and sent motivational educational materials related to gestational diabetes. Platform acts as a channel for support by physicians and for peer support | NR | Doctor, nurse | NR |
| Liu et al, 2018 [32] (China) | Age: 27.8 years, gestational age 30.3 weeks, Total participants: 98 | NR | Web based platform which patients send SMBG, for review and feedback. Feedback include advice on diet, exercise and diabetes related problem | NR | NR | Routine care, SMBG and diabetes education |

(*Continued*)

**Table 2.** (Continued)

| Author, Year (Country) | Participants characteristics* | Definition of GDM | Intervention component details | Monitoring frequency in study | Diabetes care team | Comparator |
|---|---|---|---|---|---|---|
| Miremberg et al, 2018 [28] (Israel) | Age: 31.9 years, gestational age <34 weeks, BMI: 27.1 kg/m² (5.2), Total participants: 120 | 50-g glucose challenge, and 100 g OGTT with two or more results of fasting >95 mg/dL, 1-h ≥180 mg/dL, 2-h ≥155 mg/dL, or 3-h ≥140 mg/dL | SMBG daily sent via mobile app (Glucose Buddy) for review, with feedback sent daily. Feedback include positive messaging, dietary advice, medication modification as well as advice on diabetes related problem. | Four times daily, fasting, and three 2-hr post-prandial | Maternal-foetal specialist, dietician, physician | Regular SMBG monitoring as per intervention recorded manually into diary for review, education, dietary and physical activity counselling |
| Mackillop et al, 2018 [41] (United Kingdom) | Age: 33.5 years, gestational age 24–28 weeks, BMI: 31.4 kg/m², Total participants: 203 | International Association of Diabetes and Pregnancy Study Group criteria | Mobile phone preinstalled with GDm-health app, which was uploaded and reviewed by a research midwife three times a week. Team sent SMS message on dietary advice, medication dose adjustments and counselling. | Four times daily for 3 days of the week, fasting, and three 2-hr post-prandial | Midwife | Regular SMBG as per intervention group, results recorded into logbook, which was reviewed at prenatal visits every 2–4 weeks. |
| Rasekaba et al, 2018 [49] (Australia) | Age: 32 years (5), gestational age 28 weeks (5), Total participants: 95 | International Association of Diabetes and Pregnancy Study Group criteria | Web based platform which patients send SMBG, compliance to dietary treatment and symptoms daily, with SMS reminder sent in the event no data is transmitted. Healthcare provider review data and provide counselling and feedback within one or two days. | Four times daily, fasting, and three 2-hr post-prandial | Diabetic nurse, dietitians, endocrinologist | Regular SMBG monitoring recorded onto diary and dietary advice |
| Weng et al, 2018 [31] (China) | Age: 39 years, gestational age 36 weeks, Total participants: 120 | 75 g OGTT results more than: fasting >5.1mmol/l, 2-h ≥8.5 mmol/L or two FPG readings of >5.8 mmol/L | Social media app (WeChat) which patients are sent educational materials related to gestational diabetes and a personalised nutritional and diet plan. Patients also participate in an online peer support group | NR | Diabetic nurse | Routine care including diabetes education and nutrition |
| Zhao et al, 2018 [30] (China) | Total participants: 60 | International Association of Diabetes and Pregnancy Study Group criteria | Mobile phone platform which patients send SMBG, for review and feedback. Feedback include positive messaging as well as advice on diabetes related problem. | NR | Obstetrician, nutritionist, diabetic nurse | Routine care and group diabetes education |
| Caballero Ruiz et al, 2017 [42] (Spain) | Total participants:90 | NR | Web based platform (Sinedie) which patients send SMBG, compliance to dietary treatment and ketonuria results every 3 days, with SMS reminder sent in the event no data is transmitted. Automated patient specific recommendation are sent on therapy change and dietary advice. | Four times daily, fasting, and three 2-hr post-prandial | Physician, endocrinologist | Regular SMBG monitoring and dietary advice |

(*Continued*)

**Table 2.** (Continued)

| Author, Year (Country) | Participants characteristics* | Definition of GDM | Intervention component details | Monitoring frequency in study | Diabetes care team | Comparator |
|---|---|---|---|---|---|---|
| Zeng et al, 2017 [34] (China) | Total participants: 86 | International Association of Diabetes and Pregnancy Study Group criteria | Social media app (WeChat) which patients are sent educational materials related to gestational diabetes as well as positive motivation. Support group meet-up with participants monthly | NR | Obstetrician, endocrinologist doctor, nutritionist, psychologist | Routine care and diabetes education during clinic visits |
| Bromuri et al, 2016 [43] (Switzerland) | Age: 32 years (5), gestational age 29.1 weeks (1.9), BMI: 30.0 kg/m$^2$ (6.9), Total participants: 24 | NR | Patient electronic portal which patients send blood glucose and medication related information which could be reviewed by the physician and feedback provided in between clinic visits. | Six times daily, fasting, two pre-prandial and three post-prandial | NR | Monitoring protocol as per intervention, with weekly or fortnightly clinic visits |
| Jiang et al, 2016 [42] (China) | Age: 25 years, gestational age 22.8 weeks, BMI: 20.6 kg/m$^2$, Total participants: 120 | NR | Social media app (WeChat) where patients sent data for review weekly by nurse. Recommendations on advice on diet and exercise are communicated with opportunities for clarification of queries | NR | Diabetic nurse, Nutritionist | Routine care and diabetes education during clinic visits |
| Bartholomew et al, 2015 [58] (United States) | Age: 33.2 years (5.4), gestational age 23.8 weeks (6.0), Total participants: 74 | Carpenter and Coustan criteria | SMBG sent via cell-phone internet technology system for review by physicians. All recommendations were communicated to patient via nurse. Feedback was provided include positive messaging, dietary advice, medication modification | Four times daily, fasting and three 2-hr post-prandial | Foetal medicine physician, nurse | Regular SMBG recorded into logbook and reported to nurse weekly using voicemail. Physician will review the results and make recommendations which is conveyed by the nurse |
| Khorshidi Roozbahani et al, 2015 [38] (Iran) | Age: 30.8 years (5.1), gestational age 24–28 weeks, BMI: 34.1 kg/m$^2$ (9.2), Total participants: 80 | NR | Fortnightly telephone calls from week 28–36 gestation and weekly thereafter until week 38. During calls, counselling on insulin doses, dietary advice and diabetes-related problems. Average call duration was 10 to 15 mins. | Five times daily, in the morning before breakfast, at bedtime and three 2-hr post-prandial | NR | Three telephone call at weeks 28, 32 and 36 to record blood sugar levels but no consultation was provided. |
| Given et al, 2015 [56] (United Kingdom) | Age: 31.7 years, gestational age 24–28 weeks, BMI: 33.1 kg/m$^2$, Total participants: 50 | 75g OGTT <7.0 mmol/l and a 2 hr glucose > 7.9 mmol/l | Telemedicine hub whereby patient sent clinical readings weekly to a central server for review by health care provider. Healthcare provider reviewed data and provided counselling and feedback within one or two days. | Up to seven times daily pre and post meals. | NR | Routine care as per National Institute of Clinical Excellence guideline, Regular SMBG as per intervention group, and fortnightly clinic visit |

(*Continued*)

**Table 2.** (Continued)

| Author, Year (Country) | Participants characteristics* | Definition of GDM | Intervention component details | Monitoring frequency in study | Diabetes care team | Comparator |
|---|---|---|---|---|---|---|
| Homko et al, 2012 [59] (United States) | Age: 30.1 years, gestational age <33 weeks, BMI: 34.1 kg/m² (9.2), Total participants: 80 | Carpenter and Coustan criteria | Web-based diabetes management system, which allowed for recording of insulin doses and hypoglycaemic episodes. Feedback, emotional support, and reinforcement regarding diabetes self-management was provided weekly. Interface also had links to educational resources in addition to standard care | Up to four times daily, in the morning before breakfast and 2-hr post-prandial, foetal movement counting and hypoglycaemic episodes | Maternal–foetal medicine subspecialists, residents, certified diabetes educators, and nutritionists | Regular SMBG as per intervention group, results recorded into logbook, which was reviewed at prenatal visits. |
| Pérez-Ferre et al, 2010 [57] (Spain) | Gestational age: <28 weeks, Total participants: 97 | Carpenter and Coustan criteria | Mobile phone preinstalled with application to allow for the transmission of SMBG values to the central database through SMS. Team reviewed patient data weekly and provided advice via SMS. Four face-to-face visits were scheduled until delivery. | NR | NR | Dietary counselling and SMBG, with four face-to-face visits |
| Homko et al, 2007 [60] (United States) | Age: 29.5 years, gestational age 27.6 weeks, BMI: 33.0 kg/m², Total participants: 63 | Carpenter and Coustan criteria | Web-based diabetes management system, which allowed for recording of insulin doses and hypoglycaemic episodes sent three time a week. Feedback and reinforcement regarding diabetes self-management was provided. Interface also had links to educational resources in addition to standard care | Daily glucose monitoring, foetal movement counting and hypoglycaemic episodes | Maternal–foetal medicine subspecialists, residents, certified diabetes educators, and nutritionists | Regular SMBG as per intervention group, results recorded into logbook, which was reviewed at prenatal visits. |

*Characteristics are reported as mean or median

ADA: American Diabetes Association; FPG: Fasting plasma glucose, NR: Not reported; OGTT: oral glucose tolerance test; PPG: Post-prandial glucose; SMS: short message service; SMBG: self-monitoring of blood glucose

[26,37], glucose monitoring frequency [60], or compliance to trial recommendations (e.g. dietary advice, physical activities or glucose monitoring, Table A in S1 Appendix) [45,47,53]. These values range between 66% to 85%, depending on definition used. Due to the heterogeneity of definitions, pooling was not performed.

### Risk of bias of included studies

Nine studies were judged to have a low risk of bias (Fig A in S1 Appendix), while the remaining nineteen had concerns of bias related to the study randomization process, as these were poorly described or unclear for most trials. Due to the nature of intervention, most studies did not blind participants and investigators. Nevertheless, several studies had mitigated this risk by blinding the statisticians and data analysts in their study.

## Maternal outcomes

Glycaemic control was reported in 19 studies. [26,35,37–41,43–48,50,55–57,59,60] Pooled analysis showed that women randomised to digital health experienced better glucose control, with lower fasting plasma glucose (mean difference: -0.33mmol/L; 95% CI: -0.59 to -0.07, $I^2$: 94%, p = 0.01, 17 studies, Fig 2), 2h-post-prandial glucose (-0.49 mmol/L; -0.83 to -0.15, $I^2$: 91%, p = 0.005, 13 studies, Fig 3) and HbA$_{1c}$ (-0.36%; -0.65 to -0.07, $I^2$: 95%, p:0.02, 8 studies, Fig 4) compared to routine care group at the end of the study.

## Delivery outcomes

Women randomised to digital health interventions demonstrated a smaller weight gain (mean difference: -1.81 kg, 95% CI: -3.37 to -0.25, $I^2$: 95%, p = 0.02, 6 studies) compared to routine care. In addition, there were fewer rates of foetal macrosomia (Relative risk: 0.67, 0.48 to 0.95,

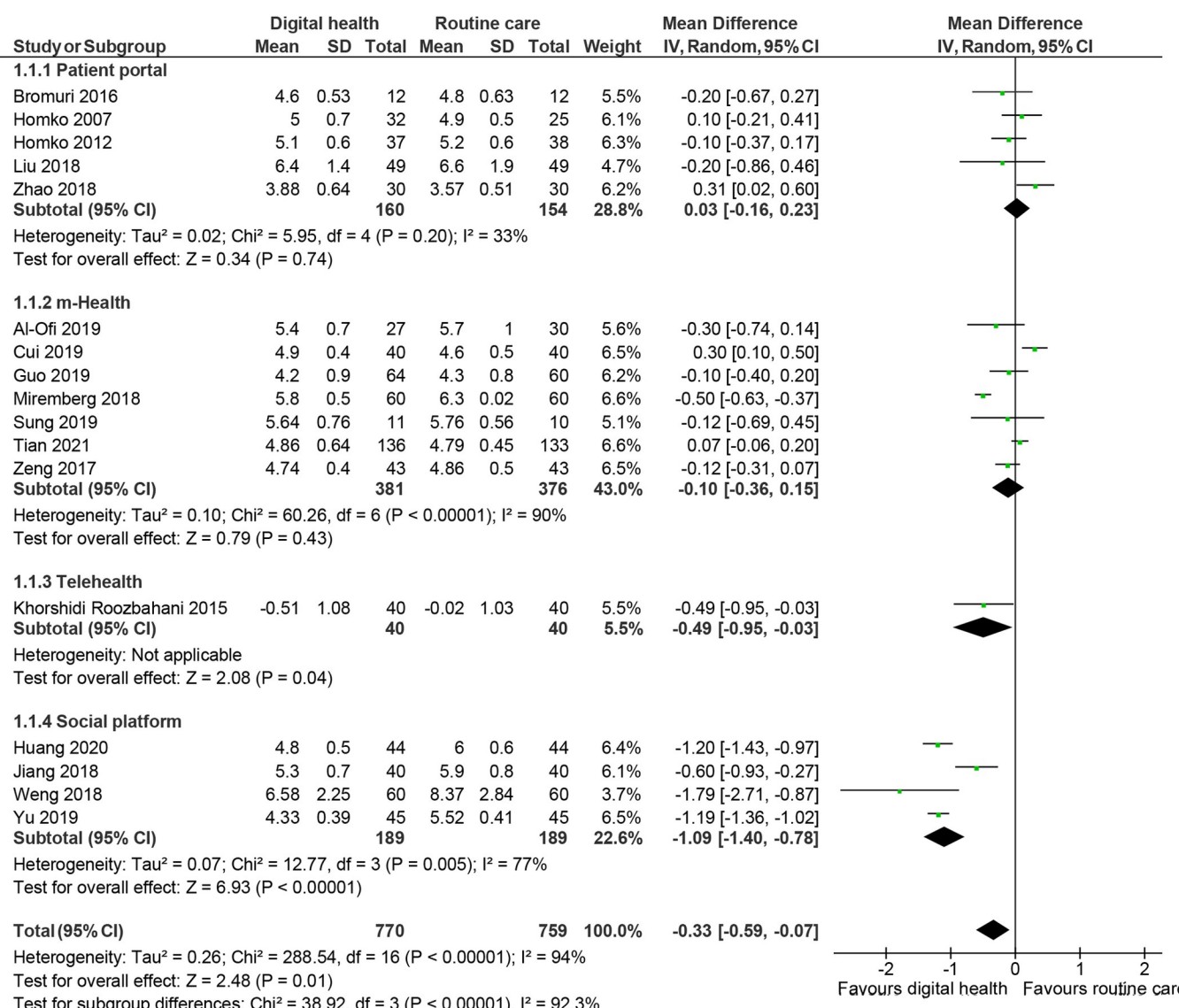

**Fig 2. Random effects meta-analysis of the mean difference in fasting plasma glucose (mmol/L), comparing digital health or routine care.**

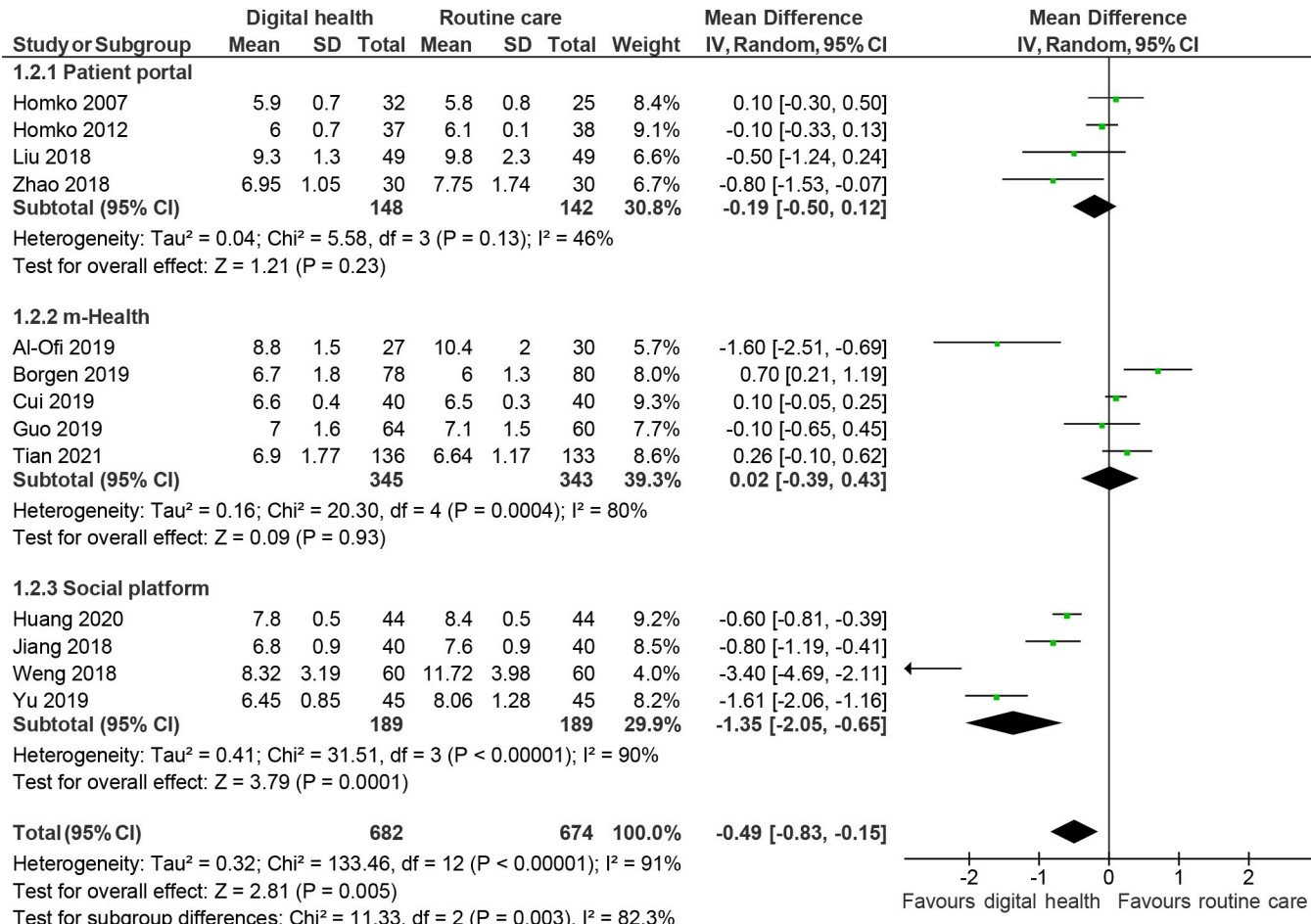

**Fig 3. Random effects meta-analysis of the mean difference in 2-hour post prandial glucose (mmol/L), comparing digital health interventions or routine care.**

$I^2$: 12%, p = 0.02, 11 studies) and caesarean delivery rates (RR: 0.81, 0.69 to 0.95, $I^2$:53%, p = 0.009). However, there was no significant difference between both groups on reducing adverse outcomes such as pre-eclampsia/eclampsia or need for use of medication (Table 3; Fig B to Fig E in S1 Appendix).

## Neonatal outcomes

Nineteen studies reported neonatal outcomes with the use of digital health interventions. [27,36–38,40,42–47,50,52,53,56,59–62] Pooled analysis showed that the use of digital health interventions was not significantly different on all neonatal outcomes examined (Table 4 & Fig F to Fig L in S1 Appendix).

## Other outcomes

Six studies reported that participants were satisfied with using digital health interventions to monitor their blood glucose, due to the ability to facilitate communication with healthcare providers. [40,41,45,54,55,58] No adverse events associated with intervention were reported. Two studies reported mental health [27,43] of patients, which did not differ between both groups while one study reported on quality of life. Two studies also examined the cost of

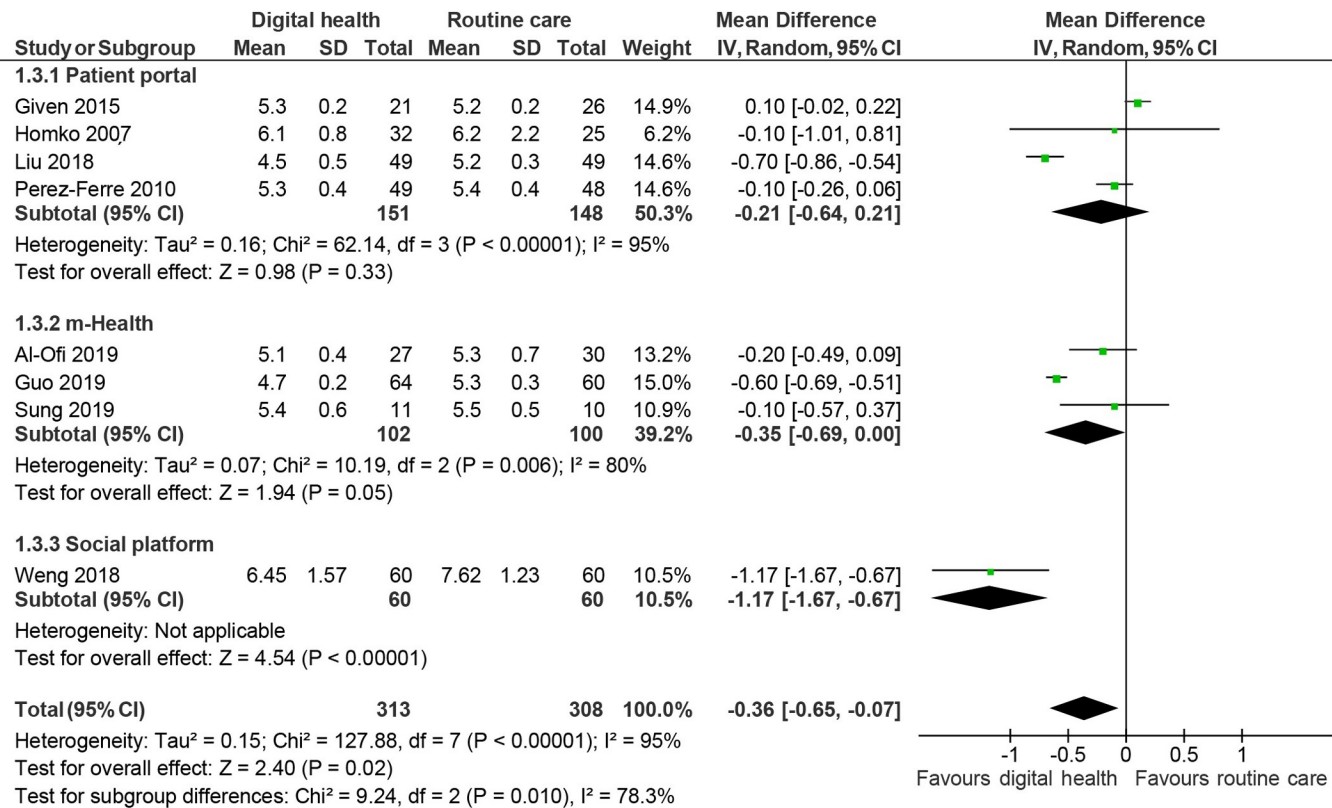

**Fig 4. Random effects meta-analysis of the mean difference in HbA1c (%), comparing digital health interventions or routine care.**

implementation of digital health interventions into the health system, which was found to be comparable between both groups. [53,61]

## Subgroup analysis

Subgroup analysis showed that the use of social platform was associated with better fasting plasma glucose control among mothers (-1.09 mmol/L; 95% CI: -1.40 to -0.78;, $I^2$: 77%, 4 studies), postprandial glucose (-1.35 mmol/L; -2.05 to -0.65, $I^2$: 90%, 4 studies) and HbA1c (-1.17%, -1.67 to -0.67, 1 study) compared to routine care at the end of study. In addition, there was lower rates of caesarean delivery (Risk ratio: 0.52, 0.38 to 0.72, $I^2$: 0%, 4 studies) and weight gain (-4.71kg; -5.46 to -3.96, 1 study) throughout pregnancy compared to routine care (Table 3). Social platform use was also found to reduce the rates of foetal macrosomia (0.22; 0.08 to 0.60, $I^2$: 0%, 3 studies) and hypoglycaemia among new-borns (0.20; 0.05 to 0.77, $I^2$: 0%, 3 studies, Table 4). Both mHealth (0.79, 0.67 to 0.92, $I^2$: 38%, 10 studies) and social platforms (0.52; 0.38 to 0.72, $I^2$: 0%, 4 studies) were effective in reducing the rates of caesarean delivery. There were fewer neonates who required the use of the neonatal intensive care unit among women randomised to mHealth compared to routine care (0.76; 0.46 to 1.28, $I^2$: 42%, 6 studies, Table 4).

When we explored whether the type of activity delivered (either telemonitoring or tele-education) had any impact on maternal and neonatal outcomes, we found that tele-education had a larger impact on glycaemic control. This could be partly explained by the small number of studies examining this form of intervention which had reported a large effect size. To explore whether cultural differences explained our results, we also stratified studies by study location

**Table 3. Pooled analyses of maternal outcomes.**

| Outcome | Technology subgroup | N of studies | N of women | Effect estimate | $I^2$ (%) | Certainty of the evidence (GRADE) |
|---|---|---|---|---|---|---|
| **Maternal outcomes** | | | | | | |
| | | | | **Mean difference [95% CI]** | | |
| Change in fasting glucose (mmol/L) | All digital health | 17 | 1,529 | -0.33 [-0.59, -0.07] | 94 | ⊕⊕⊕○ MODERATE |
| | Patient portal | 5 | 314 | 0.03 [-0.16, 0.23] | 33 | |
| | mHealth | 7 | 757 | -0.10 [-0.36, 0.15] | 90 | |
| | Telehealth | 1 | 80 | -0.49 [-0.95, -0.03] | - | |
| | Social platform | 4 | 378 | -1.09 [-1.40, -0.78] | 77 | |
| Change in postprandial glucose (mmol/L) | All digital health | 13 | 1,356 | -0.49 [-0.83, -0.15] | 91 | ⊕⊕⊕○ MODERATE |
| | Patient portal | 4 | 290 | -0.19 [-0.50, 0.12] | 46 | |
| | mHealth | 5 | 688 | 0.02 [-0.39, 0.43] | 80 | |
| | Social platform | 4 | 378 | -1.35 [-2.05, -0.65] | 90 | |
| Change in HbA$_{1c}$ (%) | All digital health | 8 | 621 | -0.36 [-0.65, -0.07] | 95 | ⊕⊕⊕○ MODERATE |
| | Patient portal | 4 | 299 | -0.21 [-0.64, 0.21] | 95 | |
| | mHealth | 3 | 202 | -0.35 [-0.69, 0.00] | 80 | |
| | Social platform | 1 | 120 | -1.17 [-1.67, -0.67] | - | |
| Weight gain over pregnancy (kg) | All digital health | 6 | 742 | -1.81 [-3.37, -0.25] | 95 | ⊕⊕○○ LOW |
| | mHealth | 4 | 572 | - 1.33 [-2.71, 0.04] | 88 | |
| | Telehealth | 1 | 80 | -0.10 [-2.19, 1.99] | - | |
| | Social platform | 1 | 90 | -4.71 [-5.46, -3.96] | - | |
| | | | | **RR [95% CI]** | | |
| Caesarean delivery rates | All digital health | 19 | 2,511 | 0.81 [0.69, 0.95] | 53 | ⊕⊕⊕⊕ HIGH |
| | Patient portal | 5 | 371 | 1.24 [0.90, 1.70] | 34 | |
| | mHealth | 10 | 1,762 | 0.79 [0.67, 0.92] | 38 | |
| | Social platform | 4 | 378 | 0.52 [0.38, 0.72] | 0 | |
| Incidence of preeclampsia/ eclampsia | All digital health | 6 | 590 | 0.81 [0.36, 1.82] | 23 | ⊕⊕○○ LOW |
| | Patient portal | 3 | 179 | 1.32 [0.59, 2.98] | 0 | |
| | mHealth | 2 | 323 | 0.60 [0.08, 4.37] | 52 | |
| | Social platform | 1 | 88 | 0.17 [0.02, 1.33] | - | |
| Use of medication | All digital health | 7 | 704 | 0.93 [0.74, 1.17] | 57 | ⊕⊕○○ LOW |
| | Patient portal | 5 | 298 | 0.79 [0.59, 1.06] | 35 | |
| | mHealth | 2 | 406 | 1.14 [0.88, 1.48] | 63 | |

mHealth: mobile health; RR: relative risk

and found that studies from Asia showed significant between group differences in maternal glycaemic outcomes (Table B in S1 Appendix). We also examined if results differed when high-quality studies were only included in our analyses. Results showed that the use of digital health interventions had minimal impact on all maternal and neonatal outcomes after inclusion of only high-quality studies (Table C in S1 Appendix). Visual inspection of funnel plots did not show any obvious asymmetry and were non-significant using Egger's test (Fig M and Fig N in S1 Appendix).

## Quality of evidence

GRADE assessment of the outcomes showed that the quality of evidence was high for caesarean delivery rates and foetal macrosomia. The quality of evidence was moderate for all

**Table 4. Pooled analyses of neonatal outcomes.**

| Outcome | Technology subgroup | *N* of studies | *N* of women | Effect estimate | $I^2$ (%) | Certainty of the evidence (GRADE) |
|---|---|---|---|---|---|---|
| **Neonatal outcomes** | | | | **RR [95% CI]** | | |
| Hypoglycaemia of new-born | All digital health | 11 | 1,316 | 0.77 [0.57, 1.05] | 10 | ⊕⊕⊕◯ MODERATE |
| | Patient portal | 3 | 176 | 1.00 [0.47, 2.13] | 0 | |
| | mHealth | 5 | 852 | 0.79 [0.53, 1.19] | 35 | |
| | Social platform | 3 | 288 | 0.20 [0.05, 0.77] | 0 | |
| Preterm birth | All digital health | 11 | 1,687 | 0.79 [0.47, 1.32] | 43 | ⊕⊕⊕◯ MODERATE |
| | Patient portal | 3 | 179 | 0.68 [0.31, 1.50] | 0 | |
| | mHealth | 5 | 1,220 | 1.11 [0.51, 2.42] | 65 | |
| | Social platform | 3 | 288 | 0.38 [0.13, 1.06] | 0 | |
| Neonatal intensive care unit | All digital health | 8 | 1,304 | 0.88 [0.58, 1.33] | 36 | ⊕⊕⊕◯ MODERATE |
| | Patient portal | 2 | 102 | 1.28 [0.70, 2.34] | 0 | |
| | mHealth | 6 | 1,202 | 0.76 [0.46, 1.28] | 42 | |
| Incidence of foetal macrosomia | All digital health | 11 | 1,764 | 0. 67 [0.48, 0.95] | 12 | ⊕⊕⊕⊕ HIGH |
| | Patient portal | 2 | 142 | 0.65 [0.18, 2.32] | 5 | |
| | mHealth | 6 | 1,334 | 0.78 [0.56, 1.08] | 0 | |
| | Social platform | 3 | 288 | 0.22 [0.08, 0.60] | 0 | |
| Large for gestational age | All digital health | 5 | 444 | 1.35 [0.77, 2.38] | 0 | ⊕◯◯◯ VERY LOW |
| | Patient portal | 2 | 185 | 1.41 [0.62, 3.20] | 0 | |
| | mHealth | 3 | 259 | 1.30 [0.60, 2.83] | 0 | |
| Small for gestational age | Patient portal | 3 | 123 | 1.58 [0.64, 3.90] | 0 | ⊕◯◯◯ VERY LOW |
| | Patient portal | 2 | 104 | 1.15 [0.14, 9.25] | 49 | |
| | mHealth | 1 | 19 | 1.35 [0.29, 6.34] | - | |
| | | | | **Mean difference [95% CI]** | | |
| Infant birth weight (g) | All digital health | 10 | 1,116 | 26.58 [-43.59, 96.75] | 21 | ⊕⊕⊕◯ MODERATE |
| | Patient portal | 4 | 274 | 122.74 [6.46, 239.02] | 0 | |
| | mHealth | 5 | 762 | -4.90 [-73.34, 63.55] | 0 | |
| | Telehealth | 1 | 80 | -175.00 [-482.91, 132.91] | - | |

mHealth: mobile health; RR: relative risk

glycaemic outcomes, and some neonatal outcomes (new-born hypoglycaemia, pre-term birth, birth weight and use of intensive care). Other outcomes were rated as low to very low quality (Table D and Table E in S1 Appendix).

## Discussion

### Main findings

In this review, we noted that the use of digital health interventions could improve maternal glycaemic outcomes such as fasting plasma glucose and HbA$_{1c}$ as well as delivery outcomes including foetal macrosomia and need for caesarean delivery. In terms of adverse neonatal outcomes such as risk of hypoglycaemia and need for neonatal intensive care unit hospitalization results were not significantly different between both groups. Taken together, this systematic review and meta-analysis provides some evidence to suggest that digital health interventions could be a useful adjunct to routine care. However, before digital health interventions can be widely implemented in routine practice, clinicians will also need to take into consideration the quality of evidence found in this review, which was rated to be mostly moderate to very low, as well as the limitations of trials which were mostly very small in sample

sizes, with varying levels of care. In our subgroup analyses of only high-quality trials, these results were not significantly different from usual care.

The management of GDM is a very time-consuming activity, for both healthcare providers and patients. Digital health interventions can help in educating women, improve self-care through nutritional and exercise advice, improve monitoring adherence and possibly improve the care of women with GDM. As such, in our analysis, we combined the various types of digital health interventions used by researchers in our meta-analysis, including telehealth (providing clinical services from a distance), mHealth (use of mobile phones for remote connection and transmission of clinical data to provider for feedback), and patient platforms (using interactive platform to support and facilitate clinical care). We took this approach as we believe that the goal was similar, which was to improve GDM care. This wide variation of practice may have contributed to some of the observed heterogeneity and can be partly explained from the subgroup analyses. The meta-analysis results suggest that intervention which involves education and monitoring of mothers with GDM can lead to improved glycaemic control. This could possibly be due to these activities which increased the level of interaction between participants and healthcare providers and thus compliance to treatment advice.

Other digital health interventions that have not been thoroughly examined is the use of social platforms and game-based support in GDM. Social media sites such as Facebook or Twitter use has increased dramatically over the past few years especially among the younger population. While there is some evidence on the effects of social media on health behaviour outcomes, [63,64] our review found only limited trials that were conducted to date. Most of the studies have mainly focused on the use of health applications, perhaps due to the popularity of such applications and high smartphone ownerships worldwide. [65,66] In addition, the use of gamification has been shown to be useful to target behavioural outcomes, especially to increase physical activity and encourage medication adherence. [67] In this review, we only found one study which attempted to use Nintendo Wii to encourage physical activities among women with GDM (NCT03073551). Given the limited evidence base to support the overall use of digital health interventions to improve self-care in GDM, additional scientific inquiry and evidence is needed before it can be recommended in routine medical practice.

Poor compliance has been associated with poor glycaemic control. Given the central importance for people with GDM to self-manage, it was surprising that very few studies adequately quantified compliance of their study participants to the recommendations, with an overall compliance rate of 66% to 85%, depending on definitions used. This may stem from the lack of a standardised definition to report study compliance, unlike studies reporting on medication adherence where adherence is often defined when a person is compliant to at least 80% of prescribed therapy (Table A in S1 Appendix). [68] Given the importance of this, we recommend that future trials ensure that these data are captured and reported.

Very few trials have reported measures of participant satisfaction or well-being, which are now increasingly being recognized as important outcomes that impact health. [69] In the review, only one study reported on quality of life, while only a few studies had reported on patient and/or physician satisfaction with digital health interventions. Only two studies assessed the economic benefits and cost-effectiveness of digital health interventions, resulting in a lack of evidence that may contribute to the underfunding and shortcomings of infrastructure. [12] Importantly, there are no studies that have examined the economic viability of any digital health interventions from low-middle income countries, where the potential for digital health interventions are the biggest in terms of reducing health inequalities, by reducing geographical barriers and increasing efficiencies and convenience. [15,70]

## Strengths and limitations

This is one of the most comprehensive systematic review examining on the use of digital health interventions in GDM focusing on both maternal and foetal outcomes from 28 studies derived from different countries and regions of Asia, Australia, Europe and the United States including 3,228 pregnant women. This robust global population, from both low-middle- and high-income countries makes our finding more robust. This needs to be taken in light of the limitations of this study. Firstly, most of the included studies had some form of methodological limitations and bias, as the reporting of methods was very poor and only few studies had published the study research protocols, which precluded us from conducting further detailed analysis such as the impact of medications or even diet.

Secondly, most of the trials were pilot studies of short duration which were designed to establish feasibility or acceptability and potential effectiveness. The total number of women with GDM recruited were mostly small and for some comparisons, only included one or two centres. This could be possibly due to the difficulty in recruitment, and as such, researchers should consider alternative study designs such as step-wedge design, conducted over multi-centres and countries to maximize recruitment. These studies should adequately describe the impact of diet, oral hypoglycaemic agents or insulin on glycaemic control. There was no long-term follow-up conducted in all studies included in the review, such as the number of women who subsequently developed diabetes or the effects of GDM on offspring such as growth parameters at 2 years.

There was considerable variation in the types of digital health technology used, the level of care participants received as well as study duration. Similarly, there was variation in the type of intervention used by each trial. For example, some trials could have included elements of education and monitoring. However, to simplify our classification and reduce any chance of mis-classification, we classified the intervention based upon the primary aim of the study, depending on whether it was to provide education or to improve blood glucose monitoring. These variations in classification and GDM definition depending on study regions and countries could have contributed to the observed heterogeneity and explain the difference in results reported by these studies. In addition, we also pooled studies using different technologies such as smartphone apps, telemonitoring devices together in our analyses. While this may not be ideal, the objective of these studies was to improve glycaemic control and support patient self-efficacy. As such, we took a broader view of this and pooled the results together and stratified this in our analyses. We found that the use of social platforms and to a smaller extent mHealth had the most benefits. Finally, our search may have missed some potential studies despite our comprehensive search in seven databases.

## Comparison with existing literature

Studies and reviews performed to date have shown that digital health interventions can be used to facilitate clinical support, monitoring of care as well as capacity building. O'Brien in one of the earliest review on this topic examined how technology can be used to support lifestyle interventions in pregnant women. [71] Nevertheless, similar to our study, they noted that the evidence base were weak, due to the lack of studies as well as poor evidence base. A more focused review by Ming and colleagues comparing digital health technology in GDM found that digital health interventions resulted in modest improvement in maternal glycaemic control. [24] Similar to the study by O'Brien, the authors noted only few randomised controlled studies that have been conducted to date. A more updated review by Garg *et al* in 2020 examined the impact of mHealth use on people with GDM. [25] In their review of 11 studies, the authors reported that mHealth could be useful to enhance patient care and support self-

management. Similar to these studies, our review on a broader topic encompassing different technologies found that digital health interventions can be effective in improving maternal outcomes

## Interpretation and future direction

Our results are timely as the incidence of GDM is increasing rapidly and with the explosion of use of digital health interventions currently. This is especially relevant with the growing interest in digital technologies and the increasing affordability of such technologies to date. However, as data are only derived from a very small cohort of subjects and the very long-term consequences on both maternal and foetal outcomes are currently lacking, further research is needed before digital health interventions can be used in clinical practice. As achieving optimal outcomes in GDM requires individuals to perform complex daily self-care tasks, it is important that any future intervention designed should adequately support and sustain these activities. [9] Indeed, in many of these trials, the human-computer interaction is often overlooked, due to poor study design which lacked a theoretical framework.

In summary, despite the widespread availability and growing use of health apps and web-based portals, such technology remains understudied in GDM. The present meta-analysis showed some benefits of using digital health interventions in improving glycaemic control and neonatal outcomes. As such, additional research is needed in view of their potential effectiveness. Indeed, the availability of a remote option provides the women and clinician to have an assessment in circumstances when they are unable to return for a scheduled appointment, as well as cost-savings to both parties due fewer clinic visits as well as tests/scans. Nevertheless, additional studies are needed which should ideally examine longitudinal outcomes to provide sufficient evidence before digital health interventions can be routinely incorporated to replace clinic visits into medical practice in GDM.

## Supporting information

**S1 Appendix. Search strategy used in this study for each database.**
(DOCX)

**S1 PRISMA Checklist. PRISMA Checklist.**
(DOCX)

**S1 PRISMA Abstract Checklist. PRISMA Abstract Checklist.**
(DOCX)

## Author Contributions

**Conceptualization:** Shaun Wen Huey Lee.

**Data curation:** Boutheina Leblalta, Hanane Kebaili, Ruth Sim.

**Formal analysis:** Ruth Sim, Shaun Wen Huey Lee.

**Investigation:** Boutheina Leblalta, Hanane Kebaili, Ruth Sim.

**Methodology:** Boutheina Leblalta, Hanane Kebaili, Ruth Sim, Shaun Wen Huey Lee.

**Project administration:** Shaun Wen Huey Lee.

**Supervision:** Shaun Wen Huey Lee.

**Validation:** Shaun Wen Huey Lee.

**Writing – original draft:** Boutheina Leblalta, Hanane Kebaili, Ruth Sim, Shaun Wen Huey Lee.

**Writing – review & editing:** Ruth Sim, Shaun Wen Huey Lee.

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
