## [Decision Letter · Decision Letter 0]

23 Aug 2021

PDIG-D-21-00005Digital health use in gestational diabetes mellitus on maternal and neonatal outcomes: A systematic review and meta-analysis of randomised controlled trials

PLOS Digital Health

Dear Dr. Lee,

Thank you for submitting your manuscript to PLOS Digital Health. The manuscript was reviewed by two reviewers, with methodological expertise in systematic reviews (Reviewer 1) and expertise in maternal and child health/digital health (Reviewer 2). Both reviewers agree the topic is important, the study rigorous and well presented. I also read the paper and agree with the reviewers that it is potentially a valuable contribution to the digital health literature. We have identified some areas for improvement that need attention, before the manuscript can be considered for publication by PLOS Digital Health. We invite you to submit a revised version of the manuscript that addresses the points raised during the review process. The changes we strongly recommend for acceptance are:   - restructuring the paper, placing the methods section after the introduction/before the findings;- providing further details on previous systematic reviews, as recommended by both reviewers;- providing additional analysis or comments on participants' adherence. Both reviewers comment on this, e.g. Reviewer 2 asks about studies that did not report frequency/engagement in glucose monitoring.  - explaining the different digital health interventions included in the studies, e.g. similarities and differences between mhealth, apps, social platforms,.., also in light of both reviewers' comments.

Please also copy-edit the manuscript, checking grammar and typos. 

We look forward to receiving your revised manuscript.

Kind regards,

Valentina Lichtner

Academic Editor

PLOS Digital Health

Journal Requirements:

1. We have noticed that you have cited 4 in the manuscript file. However, there are no corresponding files uploaded to the submission. Please ensure that all files are present to ensure that your paper is fully reviewed.

2. Please provide a detailed Financial Disclosure statement. This is published with the article, therefore should be completed in full sentences and contain the exact wording you wish to be published.

 Please include all sources of funding (financial or material support) for your study. List the grants (with grant number) or organizations (with url) that supported your study, including funding received from your institution. 

ii). State the initials, alongside each funding source, of each author to receive each grant.

iii). State what role the funders took in the study. If the funders had no role in your study, please state: “The funders had no role in study design, data collection and analysis, decision to publish, or preparation of the manuscript.”

iv). If any authors received a salary from any of your funders, please state which authors and which funders.

Reviewers' comments:

Reviewer's Responses to Questions

**Comments to the Author**

1. Does this manuscript meet PLOS Digital Health’s publication criteria? Is the manuscript technically sound, and do the data support the conclusions? The manuscript must describe methodologically and ethically rigorous research with conclusions that are appropriately drawn based on the data presented.

Reviewer #1: Yes

Reviewer #2: Yes

2. Has the statistical analysis been performed appropriately and rigorously?

Reviewer #1: Yes

Reviewer #2: Yes

3. Have the authors made all data underlying the findings in their manuscript fully available (please refer to the Data Availability Statement at the start of the manuscript PDF file)?

Reviewer #1: Yes

Reviewer #2: Yes

4. Is the manuscript presented in an intelligible fashion and written in standard English?

Reviewer #1: Yes

Reviewer #2: Yes

5. Review Comments to the Author

Reviewer #1: I thank the author for this timely investigation into the use of technologies for the management of GDM among pregnant women. I found the article well written, with results well presented, though with the occasional typo and the methods section in the wrong location. I have two main and two minor comments.

The review claims to have been performed to update existing findings from three cited reviews (referred to as "several" reviews). However, we are not presented with the results of these reviews, nor the studies included. Just that there was "limited support". At a minimum it would be good to know the year of publication and types of interventions included along with number of studies. At best we could read actual statistical results. This could give some indication of the number or types of studies that were reviewed here and not before.

Adherence to the interventions is an absolute key variable when talking about non-face-to-face interventions. This factor is only briefly mentioned with a single line informing the reader that adherence was at 66-85% across studies. I would have thought this to warrant greater attention, and potentially a sensitivity analysis. At a minimum this is a topic to be included in the discussion.

The study quality was assessed and is clearly an issue. Would the authors consider highlighting the few very good quality studies and their findings? Or a sensitivity analysis by quality for the results?

The included interventions are referred to as "digital health" but are clearly very varied. There should be some discussion of their current relevance. Clearly there is a shift toward the use of apps but are earlier interventions still used, any fading away?

Reviewer #2: OVERALL

• This is important and timely research given the increasing public health importance of GDM and the rise of digital technologies in maternal health.

• This statement late in the article very nicely sums up the contribution of this article: "This review suggest that the use of digital health can be a useful adjunct to supplement the care of pregnant women with GDM, as they result in similar maternal and neonatal outcomes without any adverse outcomes."

• To the point about timeliness, 9 of the 26 included studies were published after 2019. Further 16/26 were published since 2018, and 23/26 were published since 2015. This speaks to the need for this type of a systematic review and meta-analysis now. The authors may consider including a statement - or even a figure - that shows how much of this systematic review is focused on just the last few years.

CONTEXT

There were a few places in the article where more context would be helpful:

• A citation indicating that GDM is the most common medical complication of pregnancy

• Is there any data the authors can include that help the reader understand the global year-over-year increase in GDM? This is alluded to on line 159 on p24.

• Were there any policies around diabetes that could have been discussed to give a better scope of the problem?

• The authors could have gone into a bit more detail about why they specifically think that this work can reduce the need for women to advocate for themselves to specialist care.

INCLUSION

Various technologies and use cases were pooled together for this systematic review and meta-analysis. Ten studies of the 26 indicate NR for monitoring frequency. It was unclear to me if this means they did not report frequency or that they were not engaged in regular glucose monitoring. Was there a distinction between monitoring and purely educational interventions? Many in the research community talk about digital health as a single intervention, but there is more nuance to it. We are increasingly understanding this. This is not a suggestion to change the analysis, but rather to provide some clarity on the reasoning for including such a diverse set of digital health interventions.

INTERPRETATION

"The only digital health that was significantly associated with additional improved glycaemic control was the use of social platform and mHealth. " This is a key finding, perhaps *the* key finding. But it was not discussed. Please discuss why this might be and perhaps look to the reviewed articles for insight to share with the reader. This will have implications for both further research and practice.

OTHER SYSTEMATIC REVIEWS

I would like to see the authors speak to the other systematic reviews in this space: Ming et al. JMIR 2016; Garg et al. JDST 2020; O'Brien et al. EJCN 2014. The authors mention Ming, but do not discuss that article in a substantive way. The other two articles are not mentioned. (I am not an author on any of these articles, nor do I have any affiliations with them.) It is important to give the reader the full context because they may not know this literature. It's also important to acknowledge the work of others, even if there are limitations in the work. See my earlier comment about the timeliness of this review, and the recency of included studies.

MINOR EDITS

• There were a dozen or so grammatical issues throughout. That is fully to be expected on a manuscript of this length. We have captured some of them in this section. We suggest a couple copy-editing read-throughs before publication.

• throughout: be consistent with m-health, mHealth. define on first use.

• cover page: “A high certainty of evidence…“ This statement in the abstract was unclear. This is also on lines 45-47 on p5.

• p4: lines 78-79 be more explicit about why we need to reduce the need for women to utilize step-up care.

• p18: line 13 change "participant" to "participants"

• p18: lines 33-34 "we found that tele-education had a larger impact on glycaemic control, but this was mainly due to the small number of studies examining this form of intervention." -- this statement requires further explanation

• p24: line 170 explain what “inappropriate self care behavior” looks like

6. PLOS authors have the option to publish the peer review history of their article (what does this mean?). If published, this will include your full peer review and any attached files.

**Do you want your identity to be public for this peer review?** For information about this choice, including consent withdrawal, please see our Privacy Policy.

Reviewer #1: No

Reviewer #2: No

---

## [Editor Report · Decision Letter 1]

8 Nov 2021

PDIG-D-21-00005R1

Digital health use in gestational diabetes mellitus on maternal and neonatal outcomes: A systematic review and meta-analysis of randomised controlled trials

PLOS Digital Health

Dear Dr. Lee,

Thank you for submitting your manuscript to PLOS Digital Health. After careful consideration, we feel that it has merit but does not fully meet PLOS Digital Health’s publication criteria as it currently stands. Therefore, we invite you to submit a revised version of the manuscript that addresses the points raised during the review process.

We look forward to receiving your revised manuscript.

Kind regards,

Liliana Laranjo

Section Editor

PLOS Digital Health

Journal Requirements:

Additional Editor Comments (if provided):

Dear authors,

After editorial review a few additional issues were detected that should be addressed before the manuscript can be accepted:

1. The definition of digital health provided in the introduction seems to focus on clinicians (e.g., “support clinical decisions”) and telemonitoring/telehealth (e.g., “The premise is that users of digital health have an increased interaction with health professionals”). Digital health is much broader than this and does not have to involve the clinician, it can solely support patients/consumers too. Suggest using a different definition and providing an appropriate reference. The definition provided in the methods is much better than the one in the introduction.

2. This sentence is missing a reference: “More importantly, digital health can help ensure that health resources are optimally utilised.”

3. Sentence is repeated: “Several reviews have suggested that there were limited benefits of using digital health in women with GDM”.

4. “Several reviews have suggested that there were limited benefits of using digital health in women with GDM”.: These reviews are specifically focused on telemedicine so the sentence should say telemedicine instead of digital health.

5. The introduction should also mention the Garg et al 2020 systematic review.

6. There is a problem with the references in this excerpt: “Several reviews have suggested that there were limited benefits of using digital health in women with GDM.[13-15] In one of the earliest systematic review on the use of telemedicine for people with GDM in 2015, Rasekaba et al in their meta-analysis of 3 studies reported that telemedicine had no beneficial impact on maternal outcomes.[14]However, an updated review by Ming and colleagues of seven randomised controlled studies reported that telemedicine was useful to reduce HbA1c but not maternal and neonatal outcomes.[13]”

7. Can you rephrase and better clarify the aim: “we conducted a systematic review and meta-analysis to determine the innovative use of technology in supporting routine care delivery in women with GDM.”

8. Please provide references to the definitions provided in Box for different digital health types. Given that several analyses focus on this variable, these definitions should be provided in the main manuscript.

9. Please clarify in the methods that interventions could be classified as involving more than one digital health type.

10. Please clarify whether an intervention could deliver both telemonitoring and teleeducation. Did you analyse multicomponent interventions?

11. This sentence needs to be rephrased, it is unclear what you mean by “only social platform”, and what you are comparing social platforms with: “Subgroup analysis showed that only social platform was associated with better glycaemic control among mothers, lower rates of caesarean delivery and weight gain throughout pregnancy.”

12. For the subgroup analysis on the social platforms please provide quantitative results and refer the readers to corresponding tables/supplements.

13. Please provide quantitative results: “Results showed that digital health had minimal impact on all maternal and neonatal outcomes after inclusion of only high-quality studies.”

14. The main findings section of the Discussion should only report findings from your study. The following needs to be deleted: “Digital health are thought to have a profound effect to enhance health services delivery, enabling new models of care and shifting the focus of the health systems towards person-centre care.[12, 63]”

15. The following is an incorrect interpretation. Just because there were no significant differences does not mean that they are similar. “In terms of adverse neonatal outcomes such as risk of hypoglycaemia and need for neonatal intensive care unit hospitalization, results were similar to routine care, suggesting that supplementing or replacing clinic face-to-face visit with digital health was possible.”

16. Rephrase to remove only, as there were two types of digital health interventions that were significantly associated with additional improved glycaemic control : “The only digital health that was significantly associated with additional improved glycaemic control was the use of social platform and mHealth.”

17. Can you clarify what you mean and rephrase: “The combination of these activities could have resulted in an increase the level of interaction and thus compliance to advice by the healthcare providers.”

18. The following reasons would make sense if these were non-randomized studies, which is not the case: “However, for other outcomes such as maternal and neonatal outcomes, these were not apparent with digital health use. We offer several possible reasons for the lack of additional benefits of using digital health on GDM. Firstly, most of the outcomes reported such as caesarean rates, admission to intensive care unit are influenced directly by local clinical practice, and may not be directly influenced by intervention. Secondly, these interventions are inherently prone to selection bias, as women with easy accessibility to electronic devices are able to participate in the trials. As these information were not reported in all studies, it would be difficult assess how applicable these results would be to the wider general population, and thus the suggested benefits are speculative.”

19. Provide references: “While there is some evidence on the effects of social media on health behaviour outcomes”

20. “As such, additional scientific inquiry and evidence is needed before any digital health can be recommended as a viable means to improve self-care in GDM.” Do you really mean “any” digital health intervention or just the ones mentioned in this paragraph. Please rephrase for clarity.

21. “This may stem from the lack of a standardised definition to report study compliance”. Can you provide the definitions used in the trials as a supplement table?

22. “We found that the use of social platforms and to a smaller extent mHealth had the most benefits, and thus should be considered the highest priority in future uptake.” Suggest deleting after the comma, too speculative.

23. “O’Brien in one of the earliest review on this topic examined how technology can be used to support lifestyle interventions in pregnant women.” Reference?

24. “most of the results were similar in terms of maternal and neonatal outcomes”: “similar” is not a correct interpretation as previously mentioned.

25. S2 Table: Can you revise to make sure the comments are correct for all outcomes (e.g. caesarean)

26. There are several typos throughout; suggest proof-reading.

27. Throughout the manuscript there are many instances where you should refer to “digital health interventions”, not just “digital health”.
---

## [Editor Report · Decision Letter 2]

5 Dec 2021

PDIG-D-21-00005R2

Digital health interventions for gestational diabetes mellitus: A systematic review and meta-analysis of randomised controlled trials

PLOS Digital Health

Dear Dr. Lee,

Thank you for submitting your manuscript to PLOS Digital Health. After careful consideration, we feel that it has merit but does not fully meet PLOS Digital Health’s publication criteria as it currently stands. Therefore, we invite you to submit a revised version of the manuscript that addresses the points raised during the review process.

Thank you for submitting a revised manuscript. There are still some minor problems that should be addressed before the paper can be accepted:

Comment #16. As mentioned in the previous comment, the authors can only say that the results were not significantly different, they cannot say they are similar.

Comment #17. The wording should be improved for clarity.

We look forward to receiving your revised manuscript.

Kind regards,

Liliana Laranjo

Section Editor

PLOS Digital Health
---

## [Editor Report · Decision Letter 3]

21 Dec 2021

Digital health interventions for gestational diabetes mellitus: A systematic review and meta-analysis of randomised controlled trials

PDIG-D-21-00005R3

Dear Dr. Lee,

We're pleased to inform you that your manuscript has been judged scientifically suitable for publication and will be formally accepted for publication once it meets all outstanding technical requirements. 

Within one week, you'll receive an e-mail detailing the required amendments. When these have been addressed, you'll receive a formal acceptance letter and your manuscript will be scheduled for publication. The journal will begin publishing content in early 2022.

An invoice for payment will follow shortly after the formal acceptance. To ensure an efficient process, please log into Editorial Manager at https://www.editorialmanager.com/pdig/ click the 'Update My Information' link at the top of the page, and double check that your user information is up-to-date. If you have any billing related questions, please contact our Author Billing department directly at authorbilling@plos.org.

Kind regards,

Valentina Lichtner

Academic Editor

PLOS Digital Health

Additional Editor Comments (optional):

The paper has been much improved through the revisions. My only remaining concern is that the Author Summary appears to be more positive about the intervention than the study findings would grant. It says: 'We found that the use of digital health interventions were associated with better glucose control and lower weight gains over pregnancy, which reduces the risk of complications for both baby and mother during delivery. We also found that mothers with gestational diabetes had lower need for caesarean delivery while their babies were more likely to be born within the recommended weight range'. This, read by itself, without reading the paper and with no disclaimers on limitations, may be interpreted more positively than intended. It would be good if the authors could close the summary by still recommending not to just rely on the technology, e.g. attend follow up clinics.